# Multicenter Retrospective Analysis of Intradiscal Condoliase Injection Therapy for Lumbar Disc Herniation

**DOI:** 10.3390/medicina58091284

**Published:** 2022-09-15

**Authors:** Yusuke Oshita, Daisuke Matsuyama, Daisuke Sakai, Jordy Schol, Eiki Shirasawa, Haruka Emori, Kazuyuki Segami, Shu Takahashi, Kazumichi Yagura, Masayuki Miyagi, Wataru Saito, Takayuki Imura, Toshiyuki Nakazawa, Gen Inoue, Akihiko Hiyama, Hiroyuki Katoh, Tsutomu Akazawa, Koji Kanzaki, Masato Sato, Masashi Takaso, Masahiko Watanabe

**Affiliations:** 1Department of Orthopaedic Surgery, Showa University Northern Yokohama Hospital, Yokohama 224-8503, Kanagawa, Japan; 2Nonprofit Organization, Kanagawa Spine Research Society, Isehara 259-1193, Kanagawa, Japan; 3Department of Orthopaedic Surgery, Hatano Red Cross Hospital, Hatano 257-0017, Kanagawa, Japan; 4Department of Orthopaedic Surgery, Tokai University School of Medicine, Isehara 259-1193, Kanagawa, Japan; 5Department of Orthopaedic Surgery, Kitasato University School of Medicine, Sagamihara 252-0374, Kanagawa, Japan; 6Department of Orthopaedic Surgery, Showa University Fujigaoka Hospital, Yokohama, 227-8501, Kanagawa, Japan; 7Department of Orthopaedic Surgery, St. Marianna University School of Medicine, Kawasaki 216-8511, Kanagawa, Japan

**Keywords:** condoliase, intervertebral injection therapy, lumbar disc herniation, minimally invasive treatment, low back pain

## Abstract

*Background and Objectives*: Intradiscal injection of Condoliase (chondroitin sulfate ABC endolyase), a glycosaminoglycan-degrading enzyme, is employed as a minimally invasive treatment for lumbar disc herniation (LDH) and represents a promising option between conservative treatment and surgical intervention. Since its 2018 approval in Japan, multiple single-site trails have highlighted its effectiveness, however, the effect of LDH types, and influences of patient age, sex, etc., on treatment success remains unclear. Moreover, data on teenagers and elderly patients has not been reported. In this retrospective multi-center study, we sought to classify prognostic factors for successful condoliase treatment for LDH and assess its effect on patients < 20 and ≥70 years old. *Materials and Methods*: We reviewed the records of 137 LDH patients treated through condoliase at four Japanese institutions and assessed its effectiveness among different age categories on alleviation of visual analog scale (VAS) of leg pain, low back pain and numbness, as well as ODI and JOA scores. Moreover, we divided them into either a “group-A” category if a ≥50% improvement in baseline leg pain VAS was observed or “group-N” if VAS leg pain improved <50%. Next, we assessed the differences in clinical and demographic distribution between group-A and group-N. *Results*: Fifty-five patients were classified as group-A (77.5%) and 16 patients were allocated to group-N (22.5%). A significant difference in Pfirrmann classification was found between both cohorts, with grade IV suggested to be most receptive. A posterior disc angle > 5° was also found to approach statical significance. In all age groups, average VAS scores showed improvement. However, 75% of adolescent patients showed deterioration in Pfirrmann classification following treatment. *Conclusions*: Intradiscal condoliase injection is an effective treatment for LDH, even in patients with large vertebral translation and posterior disc angles, regardless of age. However, since condoliase imposes a risk of progressing disc degeneration, its indication for younger patients remains controversial.

## 1. Introduction

Lumbar disc herniation (LDH) is a progressive pathology caused by genetics, excessive loading, and aging, and involves deterioration of intervertebral disc (IVD) matrix organization. This deterioration may weaken the annulus fibrosus (AF) thereby potentially resulting in AF bulging or AF rupture and extrusion of nucleus pulposus (NP) material from the IVD [1,2]. The herniated tissue might compress and inflame spinal nerve structures, thereby causing lower back and/or leg pain, which can severely restrict patient activity and wellbeing [3]. Primary recommended treatments for LDH are conservative approaches, however surgery may be considered for cases showing non-responsive to prolonged conservative treatment [4]. Moreover, surgical intervention has been proposed as a more cost-effective intervention than long-term conservative care [5], nevertheless, it comes with surgery-related risks e.g., infections and surgical complications, that might require revision surgery or generate worsened symptoms [6]. Ideally, new treatment strategies are developed that provide therapeutic options for LDH falling between conservative care and surgical interventions. Specific interest is in the development of intradiscal injection products aimed to alleviate back pain and/or resolve the damaged IVD structure [7], such as growth factors therapy [8], cell transplantation [9], or chemonucleolytic drugs [10]. The latter involves the injection of an enzymatic agent that can digest part of the IVD tissue, aiming to reduce the disc herniation size and limit the impediment of nervous structures thereby augmenting symptoms [6,11,12]. The first report on chemonucleolysis dates back to 1964 [13] involving the percutaneous injection of chymopapain B, a cysteine protease, and demonstrated overall favorable results [13,14]. Nevertheless, due to controversy, commercial reasons, and concerns regarding anaphylactic complications [15], chymopapain B has become largely disemployed in clinical practice nowadays [16].

Chondroitin sulfate ABC endolyase (Condoliase) is an mucopolysaccharide-degrading lyase that specifically cleaves glycosaminoglycans with high specificity to chondroitin sulfate and hyaluronic acid [17,18]. Unlike cysteine protease chymopapin B, condoliase, can therefore specifically degrade proteoglycan-rich NP tissue, while surrounding critical tissue structures remain largely unaffected [19]. The enzyme condoliase is isolated and purified from Gram-negative *Proteus vulgaris* bacteria [20]. As such, condoliase is considered a foreign protein and thus condoliase therapy is limited to once in a lifetime application to prevent anaphylactic reactions [6]. Its application for LDH has been approved in Japan since 2018 and has since been shown advantageous in improving clinical symptoms with general good tolerability [6,11,12,21] and has presented a high responder rate (≥50% visual analog scale (VAS) leg pain improvement) at 72.0% in the placebo-controlled trial by Chiba et al. [22] Regardless, of these promising results condoliase effectiveness for different types of herniation, disc degeneration stages, and patient characteristics remains ambiguous. Moreover, clinical trials [21,22] have thus far reported good outcomes in patients aged between 20 and 70, but have not been described in patients below the age of 20 or above the age of 70 years. Therefore, the aim of this retrospective multicenter study was to determine demographic and clinical factors that are associated with strong pain improvement (≥50% VAS leg pain) following condoliase therapy by comparing group-A (≥50% VAS leg pain improvement) and group-N (<50% improvement). Secondly, to evaluate the short-term effects of this intradiscal condoliase injection for alleviation of LDH associated symptoms in teenagers and in adults older than 70 years. Finally, we complement our study with a clinical exemplar.

## 2. Materials and Methods

This study was a retrospective multicenter case-series approved by the institutional review boards of the participating organizations. Data was collected by spine surgeons certified by the Board of the Japanese Society for Spine Surgery and Related Research, and were members of the Kanagawa Spine Research Society. Short-term outcomes were assessed via the VAS pain scores, Oswestry Disability Index (ODI) [23], as well as Japanese Orthopedic Association (JOA) scores [24].

### 2.1. Patient Record Selection and Outcome Parameters

Four Japanese institutions (Showa University Northern Yokohama Hospital, Tokai University School of Medicine Hospital, Kitasato University School of Medicine, and Showa University Fujigaoka Hospital) provided data from 137 patients who received intradiscal condoliase injection for the treatment of LDH. Specifically, patients with symptomatic LDH who wanted to avoid surgery and did not experience effective pain relief with medications or fluoroscopic nerve root block. All 137 patients were analyzed for the assessment of age on condoliase effectiveness, and categorized on age as: <20-years old (teens), 20–29 years old (20’s), 30–39 (30’s), 40–49 (40’s), 50–59 (50’s), 60–69 (60’s) and all aged 70 years or older (70 and above). For the group-A and group-N analysis, records with recorded data on initial leg pain VAS scores and at a 3-month course follow-up were included. Next, patients with ≥50% improvement in baseline leg pain VAS scores at the 3 months follow-up were defined as group-A and those who didn’t were categorized as group-N [6]. Records reporting VAS scores in cm were translated to mm by a ten-fold multiplication. Clinical features and radiological findings were compared between these groups. The effect of age, sex, body mass index (BMI), disc height, spondylolisthesis, presence of >5° posterior IVD angle and >3 mm vertebral translation, Pfirrmann classification [25], and herniation type (i.e., Protruding, subligamentous extrusion, transligamentous extrusion, and sequestration [26]), on the success of intradiscal condoliase injection on pain alleviation was evaluated. 

### 2.2. Statistical Analysis

Statistical analysis was performed using Prism 9 for MacOS (v9.4.0, GraphPad Software LLC., San Diego, CA, USA). Categorical demographic and clinical features distributions were analyzed through two-tailed Chi-square test. Continuous outcomes were confirmed normally distributed through Shapiro-Wilk test and analyzed using two-tailed unpaired *t*-test. If not parametric, a two-tailed Mann-Whitney test was employed. Finally, temporal data was assessed through Two-Way ANOVA with Geisser-Greenhouse correction, followed by post-hoc Tukey’s test (>2 categories) for multiple comparison or Sisak’s multiple comparison test (2 categories). Data is presented as means ± standard deviation. The significance level was set at *p* < 0.05. Graph plots were created by Prism 9 for MacOS (v9.4.0, GraphPad Software LLC., San Diego, CA, USA) and illustrated using Adobe Illustrator (v26.3.1, Adobe, San Jose, CA, USA).

## 3. Results

### 3.1. Group-A Versus Group-N

Records of 137 LDH patients treated with condoliase were obtained from the four participating institutions. Records with initial VAS scores for leg pain were found for 89 patients of which 71 patients also had recorded postoperative scores at 3 months follow-up. (Figure 1) The 71 patients were further categorized based on reaching 50% or more reduction in VAS leg pain scores. Specifically, group-A with 55 (77.5%) patients and group-N with 16 (22.5%) patients. (Figure 1, Table 1).

Baseline VAS leg pain scores were determined at 67.3 ± 21.5 mm for the group-A and 59.8 ± 33.8 mm in group-N (*p* = 0.606), resulting in a ≥50% improvement for the group-A with a decrease in VAS score of 56.8 ± 19.9 mm at 3 months, compared to only 12.1 ± 18.7 mm in group-N. (*p* < 0.001, Table 2) Similarly strong decrease in low back pain and numbness VAS scores as well as ODI and JOA scores were observed in the group-A cohort, with only limited improvement in group-N (Table 2).

Of the patients in group-A, 8 had protrusions, 34 had subligamentous extrusions, and 13 had transligamentous extrusions (Table 1). Of those in group-N, 4 had protrusions, 10 had subligamentous extrusions, and 2 had transligamentous extrusions. The differences in patient numbers according to herniation types were determined not statistically significant different (*p* = 0.465). Combining both cohorts, however, did suggest patients with transligamentous extrusion LDH to present higher low back pain reduction at 1 month follow-up, resulting in a significant difference between subligamentous extrusions and transligamentous extrusions (*p* = 0.015). However, overall pain reductions were similar for all types at 3 months follow-up. (Figure 2B) Of the patients in group-A; 4 had grade II disc Pfirrmann classification, 20 had grade III, 28 had grade IV, and 3 had grade V classification; of those in group-N, 11 patients had grade III, 3 had grade IV, and 2 had grade V classification. Statistical analysis revealed a significant difference (*p* = 0.047) in this Pfirrmann grade distribution, with group-A presenting higher rates (50.9%) of grade IV, while group-N primarily presented with grade III (68.8%). Combining the results of both cohorts, did show a trend of reduced improvement in pain scores, for patients presenting a Pfirrmann classification grade III, unable to report a significant improvement at 3 months in VAS low back pain. Moreover, grade IV patients reported a significantly higher VAS low back pain improvement compared to grade III (*p* = 0.048) and grade V (*p* = 0.048) patients (Figure 2C,D). Unsurprisingly, patients with Pfirrmann classification V had a worse baseline pain levels and follow-up pain scores, although improvement rates were similar to other classifications, the improvement only approached statistical significance at 3 months follow up. No differences were found in BMI, disc height, or age distributions between the two cohorts (Table 1).

High intensity zones (HIZ) were recorded in 7 (12.7%) of group-A patients and 4 (25.0%) in group-N. The presence of HIZ did not show a relation with effectiveness of the treatment (*p* = 0.232, Table 1). Similarly, collective assessment also showed similar rates of leg and low back pain VAS score improvement (Figure 2E–H). Vertebral translations of more than 3 mm was reported for 3 patients, 2 in group-A and 1 in group-N (Table 1). The distributions were not significantly different (*p* = 0.647). Collective data, however, did suggest a trend of enhanced improvement rates for patients with a >3 mm vertebral translation (Figure 2I–L). Comparing between group-A and group-N for the distribution of patients with a posterior IVD angle larger than 5° revealed a difference that approached statistical significance (*p* = 0.066). Only group-A presented this clinical feature with 10 of the 55 (18.2%) patients (Table 1), and showed an enhanced trend for VAS leg pain improvement (Figure 2M,N), as well as an initially higher improvement at the 1-month follow-up for low back pain scores (Figure 2O,P).

### 3.2. Effect of Age on Condoliase Efficacy

A total of 137 LDH patient records were retrieved (Figure 1) of which 81 were male and 56 were female. Their average age was 43.2 years (range: 16–88 years), and their average BMI was 24.3 (range: 16.1–50.3). The most common level of LDH was L4–L5. The types of herniation were protrusion in 20 patients, subligamentous extrusion in 93 patients, and transligamentous in 23 patients. With regards to Pfirrmann classification, no patients presented a grade I classification, 6 had a grade II classification, 69 had a grade III, 56 had a grade IV, and 6 had a grade V classification. Fifteen patients exhibited HIZ on T2-weighted magnetic resonance imaging (MRI) images. Four had previously undergone surgery at the same disc level. Four patients had spondylolisthesis, 3 had a vertebral translation > 3 mm, and 18 had a disc instability. Previous studies did not analyze the effect of condoliase on patients older than 70 years or below the age of 20. Our retrospective cohorts of 137 patients included 13 patients of 70 years or older and included 8 patients within the teen category (Figure 1).

Analysis of pain and disability outcomes per analysis suggested that regardless of age, an evident reduction in leg, low back pain, and numbness VAS scores could be observed (Figure 3A–C). Similarly, a clear reduction in ODI and improvement in JOA scores were observed for all age categories (Figure 3D,E).

Focusing on the teen-category (<20 years old); eight patients were younger than 20 years of which 1 had a protrusion-type hernia, 6 had subligamentous extrusion-type hernia, and 1 had a transligamentous extrusion-type hernia (Table 3). The herniations were located at L4–L5 for 5 and L5–S1 for 3 patients. None of these 8 patients had a HIZ on MRI scans. In the observation period a strong reduction in VAS low back pain and numbness scores was observed compared to the other categories (Figure 3). Most striking was the strong reduction in ODI outcomes, although records reporting ODI outcomes were few. Moreover, JOA scores reached highest improvements for the teen categories compared to other age cohorts. Although all outcomes suggested an impactful improvement (Figure 3A–E) none of the outcomes were determined statistically significant. Notably however, is that JOA scores (*p* = 0.050), VAS leg pain (*p* = 0.073), and VAS numbness (*p* = 0.051) approached statistically significant improvements at the 3-month follow-up. The lack of significance, however, is likely in part ascribed to the relatively small sample size of the cohort. On an individual level, all teen patients reported improvement in leg pain VAS scores at the 3 months follow-up, in which 2 reported full pain alleviation. (Figure 4A) Similarly, a reduction or maintenance of low baseline back pain and numbness VAS scores was reported in tracked teens at the final follow-up (Figure 4B,C). None of the patients opted to undergo surgery in our follow-up. MRI at the 3 months follow-up, however, revealed that 2 patients (25.0%) presented deterioration of Pfirrmann grade II discs to grade III and 3 (37.5%) patients presented a worsening from grade III to a grade IV (Table 3).

Concentrating on the elderly population (70 years old and above), a total of 13 patients were recognized. Here, two patients reported HIZ. Eight patients were classified as Pfirrmann grade IV, 2 as grade V, and 3 as grade II. One patient presented with protrusion LDH, 2 with transligamentous extrusion and the remaining 10 patients with subligamentous extrusion. A reduction in VAS and ODI scores and improvement in JOA scores was observed, all but VAS numbness resulting in a significant improvement at 3 months (Figure 4). Expectedly, the baseline scores indicated a higher pain and disability status prior to start of the treatment. All pain classifiers suggested overall improvement in pain following condoliase treatment, although the rate of improvement showed a slower trend than the other age categories. On an individual level, all patients reported improvement or maintenance of leg pain VAS scores at the 3-month check, with 2 patients reporting complete riddance of pain. Similarly, all but one patient reported improvement or maintenance in low back pain VAS scores, (Figure 4E) of which 2 patients reported complete resolvent at the 3 months follow-up. Numbness VAS scores showed worsening for two patients at the 3 months follow-up, while the rest improved or maintained their scores (Figure 4F).

### 3.3. Case Presentation

A 42-year-old female patient presented with L5/S1-disc herniation, classified as Pfirrmann grade IV. Three months after condoliase injection, the hernia was reduced in size, however, also a clear reduction in signal intensity was observed (Figure 5). The JOA score improved from 12 points before injection, 18 points at 1 month, and 25 points at 3 months after injection. The VAS for low back pain changed from 30 mm pre-injection, to 50 mm at 1 month, and improved to 20 mm at 3 months follow-up. Similarly, the VAS for leg pain changed from 70 mm before injection, 80 mm at 1 month, to 20 mm at 3 months follow-up. 

## 4. Discussion

According to Andersson [27], back pain is the most common cause of activity limitation in people younger than 45 years, the second most frequent reason for visits to a physician, the fifth-ranking reason for hospital admission, and the third most common reason for surgical procedures. Moreover, LDH is the primary cause of lower back pain [11]; thus, developing effective interventions for LDH is essential to improve quality of life and reduce socioeconomic healthcare costs. Previous reports have shown intradiscal condoliase injection to be safe and effective in reducing herniation size and improving patients’ symptoms [6,11,12]. Moreover, the minimally invasive procedure likely limits the impact on the patients compared to surgical intervention and is expected to reduce costs [7]. Despite these advantages, the optimal patient stratification e.g., patient age for administration of condoliase is not well known. Due to its novelty, intradiscal condoliase injection has only been described in a few reports [6,11,12] involving single-center studies. This report is the first study to examine the combined results of condoliase treatment from multiple healthcare centers. In addition, Chiba et al. and Matsuyama et al. reported that condoliase significantly improved symptoms in patients between the age of 20 and 70 years old [22], however, its impact on patients younger than 20 or older than 70 years old was previously not examined. This is a critical aspect, considering that both teenagers and senior citizens tend to favor less invasive treatments for LDH [28]. As such our clinical report presenting positive effect for both age categories may prove beneficial.

In our work we aimed to examine specific clinical- and demographic features associated with successful condoliase treatment derived pain relieve. Previous work by Okada et al. aimed at identifying prognostic factors for effective condoliase treatment found that a vertebral translation > 3 mm and a posterior disc angle > 5° may reduce efficacy of the treatment [11]. However, in our study, 3 patients with a vertebral translation > 3 mm experienced improvement, and 10 out of 18 patients with a posterior disc angle > 5° experienced improvement during their observation period. More specifically, a posterior disc angle > 5° was found at higher rates in group-A compared to the group-N pool, approaching statistical significance. Therefore, neither high vertebral translation nor posterior disc angle increased the risk of treatment failure in our study. In a similar study by Nakajima et al., responders were defined as patients presenting a ≥50% improvement in leg pain VAS scores at a three-month follow-up [6]. Under this criterion, they found 76.2% (32 of 42) of patients responded to the condoliase injection therapy. This is in line with our results, presenting 77.5% (55 of 71) of patients being classified in group-A as well as the placebo-controlled trial by Chiba et al. [22] reporting a rate of 72.0%. Of note, is that the ≥50% improvement threshold improvement in VAS leg pain, also associated with clear differences in VAS low back pain, VAS numbness, ODI, and JOA score improvement between group-N and group-A, supporting the validity of the criterium. For assessment of prognostic factors, Banno et al. reported the presence of a HIZ as a positive factor for condoliase treatment efficacy [12], however, our study couldn’t confirm these observations. A similar study by Inoue et al. [29] reported injection into disc L5/S1 as a positive factor for treatment efficacy, however, again our study reported similar proportion of L5/S1 herniation for both group-A and group-N. Notably all contemporary reported studies aimed to determine prognostic factors for effective condoliase treatment, are limited by a relatively small sample sizes and short observations periods. Our study similarly set out to find prognostic factor using a multicenter based assessment. Our analysis revealed that Pfirrmann classification might be associated with treatment efficacy. Specifically, grade III classified discs were found at much higher rates in the group-N, while grade IV were more prevalent in the group-A. This finding is in line with observations from Nakajima et al. [6] who reported a lower rate of grade IV/V discs in their group-N. However, others have reported higher rates of improvement for lower Pfirrmann classifications [11]. Whether these results might suggest condoliase treatment to present higher efficacy for more severely degeneration discs remains to be determined. Likely, larger scale studies with longer observation periods are needed to get an enhanced grasp on optimized patients’ indication for successful condoliase treatment. 

Our study is the first report to examine the effect of condoliase specifically in teenagers and elderly patients. In teenagers, all 8 cases reported improvement in leg pain, but 5 cases had worsening in Pfirrmann grades, suggesting that further consideration for the indications for younger cases is required. Moreover, condoliase appears more effective for severely degenerated discs (Pfirrmann grade IV), and since condoliase can cause degeneration of the disc [29], research into compensatory therapy to limit or reverse the condoliase induced degeneration, such as stem cell treatment [9], will likely prove beneficial. Particularly for these younger patients. In elderly patients condoliase treatment also showed able to result in clinically significant pain reductions for the large majority of patients, however, the overall improvement appeared more tempered, even compared to patients in their 60’s. Nonetheless, the impact of condoliase appears beneficial regardless of age category. 

Limitations of this study include the studies retrospective and single-arm (no data of comparison with placebo or other injection therapy) nature. Moreover, lack of information of other treatments (analgesics and physiotherapy) and condoliase treatment specifications could have affected outcome results and was not corrected for. Moreover, our and previous report have an overall limited follow-up and, despite the multicenter design sample, the included number of patients remains relatively small and heterogenous. Specifically, patient numbers in the young and elder categories were small, the made observations for these cohorts thus require examination in larger cohorts to confirm their validity. Here a follow-up period of three months was employed, as such long-term effects still remain unknown. Also, side effects and treatment complications were not analyzed in this study and remain a critical aspect to review in future work. Despite our reported improvement in pain outcomes, our study did not measure changes in the volume of herniated tissues. Further study is required to fully elucidate indications for condoliase therapy; however, initial outcomes appear very favorably is short term pain relief for LDH patients. Finally, future trials should aim to compare the safety and cost-efficacy profiles of condoliase intervention to standard microdiscectomy surgery to discern condoliase’s full potential in LDH treatment.

## 5. Conclusions

Intradiscal condoliase injection proved effective in the majority of patients with LDH, as indicated by a strong clinically-significant improvements in leg pain, low back pain, and disability outcomes. The current study suggests that patients with large vertebral translation and posterior intervertebral angles may respond to treatment, even in patients older than 70 years with advanced degenerated IVD. Considering the possibility that degeneration of the IVD may progress due to the matrix degradation effects of condoliase, the indication for younger patients remains controversial.

## Figures and Tables

**Figure 1 medicina-58-01284-f001:**
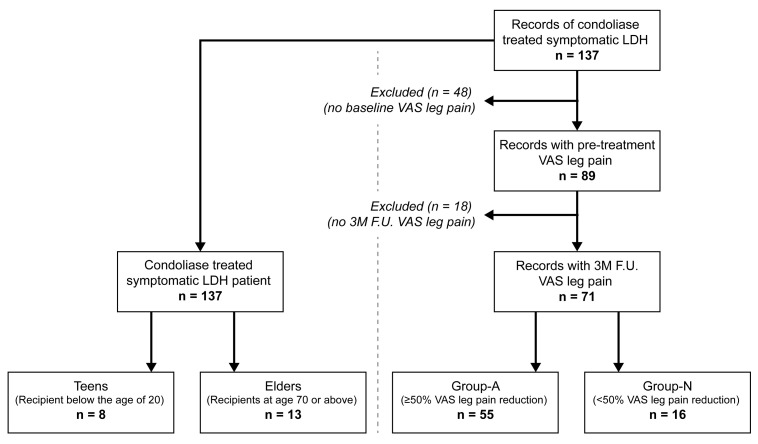
Flowchart representing identified records of lumbar disc herniation (LDH) patients treated with intradiscal condoliase treatment, and subsequent categorization into group-A and group-N and a separate categorization of teens and elders. (Abbreviations: F.U.: Follow up, LDH: lumbar disc herniation, VAS: visual analogue scale).

**Figure 2 medicina-58-01284-f002:**
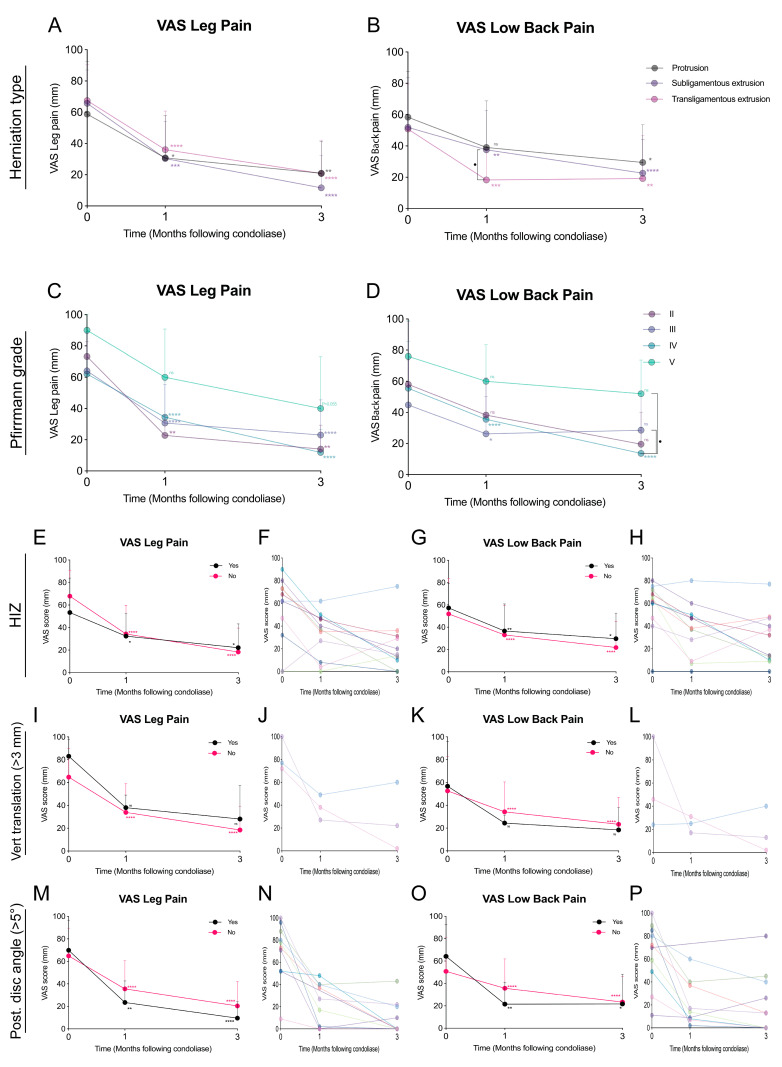
Graphical representation of the improvements in pain scores following condoliase treatment of the collective group-A and group-N cohorts analyzed for specific clinical features. The effect of herniation types on (**A**) leg pain and (**B**) low back pain visual analogue scale (VAS) scores. The effect of and Pfirrmann grades on (**C**) leg pain and (**D**) low back pain VAS scores. Comparison of patients with high intensity zones (HIZ) on (**E**,**F**) leg pain and (**G**,**H**) low back pain VAS scores, where (**F**,**H**) present scores for individual patients with HIZ. Effect of vertebral translation larger than 3 mm on (**I**,**J**) leg pain and (**K**,**L**) low back pain VAS scores, where (**J**,**L**) present scores for individual patients with the vertebral translation. Effect of disc instability as indicated by posterior disc angle larger than 5° on (**M**,**N**) leg pain and (**O**,**P**) low back pain VAS scores, where (**N**,**P**) present scores for individual patients with disc instability. Error bars represent standard deviation. Assessment within each category compared to baseline with ns: non-significant, * *p* < 0.05, ** *p* < 0.01, *** *p* < 0.005, and **** *p* < 0.001, and assessment at each time-point between conditions with • *p* < 0.05.

**Figure 3 medicina-58-01284-f003:**
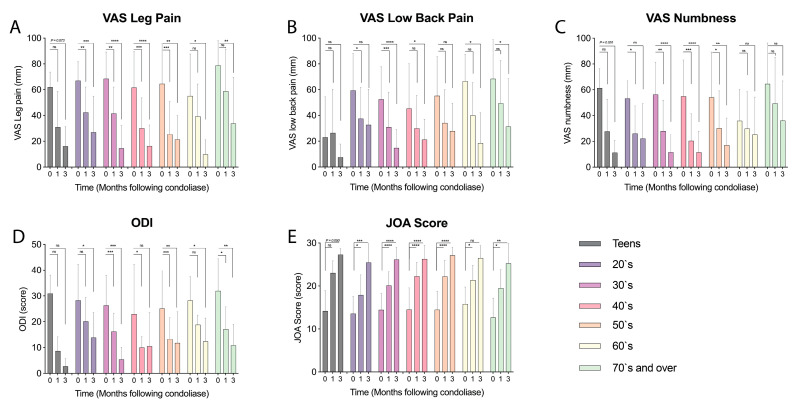
Graphical representation of changes in visual analogue scale (VAS) scores following condoliase injection for (**A**) leg pain, (**B**) low back pain, and (**C**) numbness as well as (**D**) Oswestry disability index (ODI) and (**E**) JOA scores for LDH patients categorized per age, highlighting the trends of improvement in teens and elders (70’s and over) are similar to the other age categories. Bars represent mean values and error bars standard deviation. Assessment within each category compared to baseline with * *p* < 0.05, ** *p* < 0.01, *** *p* < 0.005, and **** *p* < 0.001 or NS indicating a non-significant (*p* ≥ 0.05) difference.

**Figure 4 medicina-58-01284-f004:**
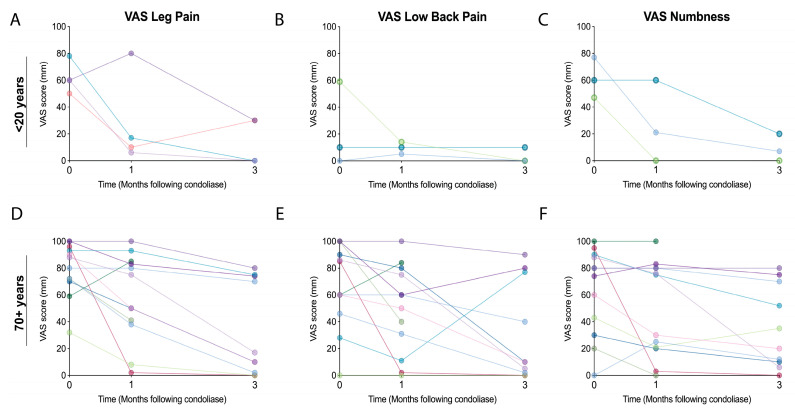
Graphical representation of individual patients in the teens ((**A**–**C**); <20 years old) or elderly ((**D**–**F**); 70 or 70+ years old) category with (**A**,**D**) leg pain, (**B**,**E**) low back pain, and (**C**,**F**) numbness visual analogue scale (VAS) scores followed for 3 months after condoliase treatment.

**Figure 5 medicina-58-01284-f005:**
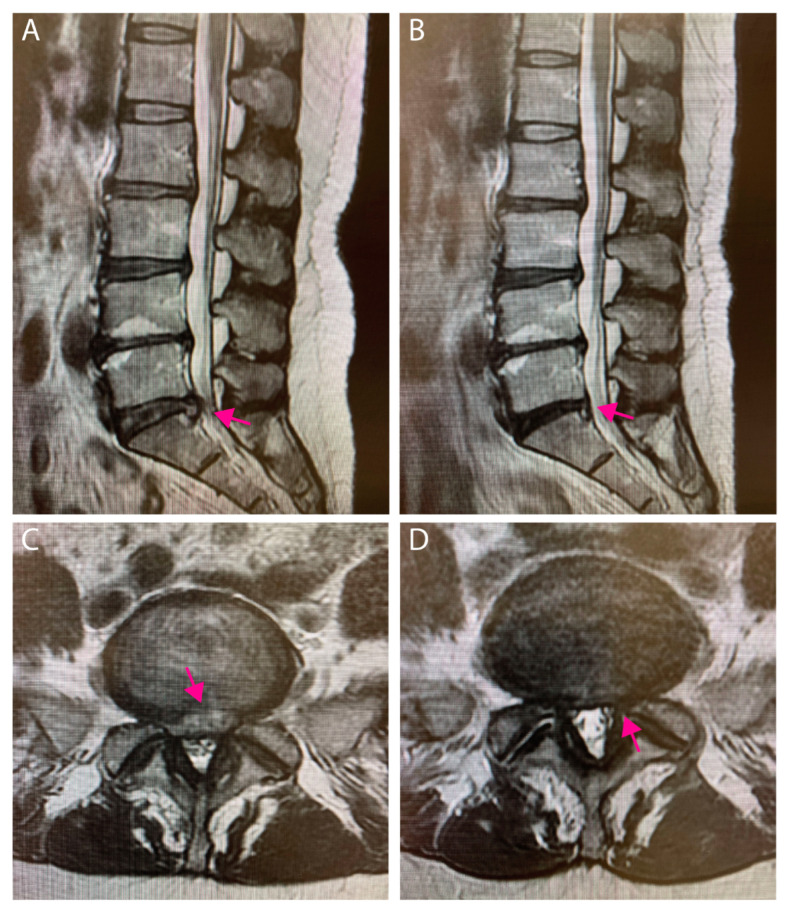
(**A**) Sagittal and (**B**) axial MRI images of same 42-year-old LDH (L5/S1) patient before condoliase injection image. (**C**) Sagittal and (**D**) axial MRI images of same patient 3 months after injection. Notice the reduction in herniation size, but also advancement of degeneration of the L5/S1 disc as indicated by loss of signal intensity. (Red arrow indicates herniated tissue).

**Table 1 medicina-58-01284-t001:** Overview of demographic and clinical features of group-A and group-N. *p*-values in bold font highlight statistically significant differences.

Parameters		Group-A		Group-N		
		*n*	%	*n*	%	*p*-Value
Number of patients		55	-	16	-	
Sex (Female)		25	(45.5%)	8	(50.0%)	0.748 ^•^
Age group	Teens	3	(5.5%)	1	(6.3%)	0.897 ^•^
	20 s	9	(16.4%)	2	(12.5%)	
	30 s	9	(16.4%)	2	(12.5%)	
	40 s	14	(25.4%)	4	(25.0%)	
	50 s	9	(16.4%)	2	(12.5%)	
	60 s	5	(9.1%)	1	(6.3%)	
	70 and over	6	(10.9%)	4	(25.0%)	
Pfirrmann classification	Grade I	0	(0%)	0	(0%)	**0.047 ^•^**
	Grade II	4	(7.3%)	0	(0%)	
	Grade III	20	(36.4%)	11	(68.8%)	
	Grade IV	28	(50.9%)	3	(18.8%)	
	Grade V	3	(5.5%)	2	(12.5%)	
Herniation type	Protrusion	8	(15.5%)	4	(25%)	0.465 ^•^
	Subligamentous extrusion	34	(61.8%)	10	(61.8%)	
	Transligamentous extrusion	13	(23.6%)	2	(12.5%)	
	Sequestration	0	(0.0%)	0	(0.0%)	
Herniation level	L2/3	1	(1.8%)	0	(0.0%)	0.759 ^•^
	L3/4	1	(1.8%)	1	(6.3%)	
	L4/5	31	(56.4%	9	(56.3%)	
	L5/S1 (or L5/6)	22	(40.0%)	6	(37.5%)	
HIZ present (yes)		7	(12.7%)	4	(25.0%)	0.232 ^•^
Spondylolisthesis (yes)		2	(3.6%)	1	(6.3%)	0.647 ^•^
Vertebral translation > 3 mm (yes)		2	(3.6%)	1	(6.3%)	0.647 ^•^
Posterior disc angle > 5° (yes)		10	(18.2%)	0	(0%)	0.066 ^•^
		**Avg ± SD**	** *n* **	**Avg ± SD**	** *n* **	***p*-Value**
Age (years)		44.8 ± 16.2	55	50.6 ± 20.3	16	0.235 ^□^
BMI		24.1 ± 6.9	44	23.4 ± 5.3	15	0.372 ^⊥^
Disc height (mm)		8.4 ± 2.2	55	9.0 ± 3.2	16	0.349 ^□^
Baseline VAS leg pain		67.3 ± 21.5	55	59.8 ± 33.8	16	0.606 ^⊥^
Baseline VAS low back pain		53.8 ± 27.0	14	52.6 ± 31.4	52	0.978 ^⊥^
Baseline VAS numbness		52.4 ± 25.5	49	52.3 ± 27.6	14	0.984 ^□^
Baseline ODI		29.4 ± 16.3	41	21.7 ± 10.6	12	0.136 ^⊥^
Baseline JOA score		14.6 ± 3.8	39	15.8 ± 3.6	13	0.537 ^⊥^

^•^—Chi-square test; ^□^—Unpaired *t*-test; ^⊥^—Mann-Whitney test; Avg—Average; BMI—Body mass index; HIZ—High-intensity zone; ODI—Oswestry disability index; SD—standard deviation; VAS—Visual analog scale.

**Table 2 medicina-58-01284-t002:** Comparison of changes compared to baseline in pain and disability outcomes between group-A and group-N at 1 month and 3 months follow-up after intradiscal condoliase injection. *p*-values determined through unpaired *t*-test, with bold font highlighting statistically significant differences.

Parameter		Group-A		Group-N		
		Avg ± SD	*n*	Avg ± SD	*n*	*p*-Value
Change VAS Leg pain (mm)	1 M	−38.1 ± 24.92	54	−10.4 ± 22.3	16	**0.002**
	3 M	−56.8 ± 19.9	55	−12.1 ± 18.7	16	**<0.001**
Change VAS LBP (mm)	1 M	−19.8 ± 26.9	51	−12.9 ± 22.4	14	0.387
	3 M	−36.4 ± 30.8	52	−6.5 ± 31.4	14	**0.002**
Change VAS numbness (mm)	1 M	−28.5 ± 29.4	48	−14.3 ± 23.5	14	0.103
	3 M	−41.0 ± 27.9	49	−14.4 ± 22.3	14	**0.002**
Change ODI	1 M	−13.9 ± 11.0	33	−3.6 ± 4.8	12	**0.003**
	3 M	−21.0 ± 12.2	40	−6.9 ± 8.3	11	**0.001**
Change JOA Score	1 M	6.9 ± 4.3	36	3.7 ± 2.9	13	**0.018**
	3 M	12.8 ± 3.7	38	7.4 ± 3.5	12	**<0.001**

1 M—1-month follow-up; 3 M—3-month follow-up; Avg—Average; LBP—Low back pain; ODI—Oswestry disability index; SD—standard deviation; VAS—Visual analog scale.

**Table 3 medicina-58-01284-t003:** Overview of demographic and clinical features of the 8 teenage (<20 years old) condoliase treated disc herniation patients and recorded changes in Pfirrmann grading.

Parameters		*n*	%
Number of patients		8	-
Age (years)	16	1	(12.5%)
	17	4	(50.0%)
	18	1	(12.5%)
	19	2	(25.0%)
Sex (Female)		2	(25.0%)
Herniation level	L4/5	5	(62.5%)
	L5/S1	3	(37.5%)
Herniation type	Protrusion	1	(12.5%)
	Subligamentous extrusion	6	(75.0%)
	Transligamentous extrusion	1	(12.5%)
	Sequestration	0	(0.0%)
High intensity zones (HIZ)		0	(0.0%)
Change Pfirrmann classification *	Grade II to III	2	(25.0%)
	Grade III to IV	3	(37.5%)
	Grade IV to unknown	1	(12.5%)
	Unchanged	2	(25.0%)

* Change in Pfirrmann classification from baseline to 3 months follow-up.

## Data Availability

The data presented in this study are available on request from the corresponding author. The data are not publicly available due to privacy and ethical restrictions.

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
