# Peer review of "Multicenter Retrospective Analysis of Intradiscal Condoliase Injection Therapy for Lumbar Disc Herniation"

_medicina, 2022, doi:10.3390/medicina58091284_

Round 1

Reviewer 1 Report

Number of teen pts is mentioned as 8 and elderly above 70 as 13 in Fig 1, table 3 and the text. But in table 1 numbers are 4 and 10 respectively. Plz check.

MRI Images in Fig 5 are not in the same plane. Also there appears to be no decrease in the size of the LDH as mentioned in the text.

The authors claim at the outset to mainly study the effect of the procedure in the teen and elderly population above 70, but most of the data and article discussion encompasses all age groups.

The number of teen and elderly pts (>70) is too small to come to any effective conclusion. Statistical significance to be ascertained.

Need more details about any side effects of the injection/ complications associated with the procedure. 

Longer follow up data if possible.

Plz check whether condoliase inj is more effective in Pfirrman Gr II & III degenerative stages as reported in some articles

If possible any comparative studies of surgery vs condoliase injection to be mentioned

Author Response

REVIEWER 1

We are very grateful for the reviewer to review our work and provide us with constructive feedback to improve our manuscript. Please find comments too all queries below. We believe the points of attention have been able to improve the quality of our work. Thank you.

1. Number of teen pts is mentioned as 8 and elderly above 70 as 13 in Fig 1, table 3 and the text. But in table 1 numbers are 4 and 10 respectively. Plz check.

These numbers are correct. Table 1 shows the elderly and younger groups consisting ONLY of patients that had both baseline and 3M follow-up VAS leg pain scores, while the aged and younger patient cohort mentioned in figure 1 and table 3, involved ALL patients in the registry regardless of baseline and FU VAS leg pain data.

2. MRI Images in Fig 5 are not in the same plane. Also there appears to be no decrease in the size of the LDH as mentioned in the text.

In response to the comments of the reviewer, we have revised the case presentation, involving a clearer MRI image showing the reabsorption of the herniated tissue after treatment. Please see the newly uploaded figure 5 and the text that reads:

“A 42-year-old female patient presented with L5/S1-disc herniation, classified as Pfirrmann grade IV. Three months after condoliase injection, the hernia was reduced in size, however, also a clear reduction in signal intensity was observed. (Figure 5) The JOA score improved from 12 points before injection, 18 points at 1 month, and 25 points at 3 months after injection. The VAS for low back pain changed from 30mm pre- injection, to 50mm at 1 month, and improved to 20mm at 3 months follow-up. Similarly, the VAS for leg pain changed from 70mm before injection, 80mm at 1 month, to 20mm at 3 months follow-up.”

3. The authors claim at the outset to mainly study the effect of the procedure in the teen and elderly population above 70, but most of the data and article discussion encompasses all age groups.

The manuscript has been separated in two `separate` assessments; one compares group-A versus group- N to examine any potential prognostic factors for successful treatment (Figure 2 and table 1-2) and the other part specifically focusses on overall outcome improvements in the younger (<20y) and older generations (70+ y). (Figure 3, and table 3-4) This aspect is illustrated in figure 1 and are separately discussed in the methods, results, and discussion and titled accordingly.

It is true, however, that we also show (in figure 3) the outcomes of other age categories. The reason for including these other age categories in the figure, was to give an indication on the overall profile of improvement seen, and how the younger and older categories compared. To emphasize this aspect, we have provided some clarifying text to the figure legend. We would however emphasize that the other age categories are not discussed in depth within the text, so we are of the opinion that our manuscript remains focused on the main aims set out in the introduction.

New line reads: “Figure 3. Graphical representation of changes in visual analogue scale (VAS) scores following condoliase injection for (A) leg pain, (B) low back pain, and (C) numbness as well as (D) Oswestry disability index (ODI) and (E) JOA scores for LDH patients categorized per age, highlighting the trends of improvement in teens and elders (70`s and over) are similar to the other age categories. ...”

4. The number of teen and elderly pts (>70) is too small to come to any effective conclusion. Statistical significance to be ascertained.

We concur with the reviewer that the cohort sizes of these age categories are relatively small and have addressed these aspects in the text accordingly. Despite this, we do believe our findings and report hold validity, in particular as our study is the first to show that the relatively new condoliase treatment shows beneficial with clinically significant effects in a majority of our 70+ year old patients; which could provide confidence for future studies to consider including patients in this category. Conversely, despite the small size and trends of improvement, the worsening in Pfirrmann classification seen in the teen (<20y) category underlines a caution for clinicians to include such young participants for condoliase intervention. We believe these two general findings are important observations to share with the scientific and clinical community.

5. Need more details about any side effects of the injection/ complications associated with the procedure.

We concur that this is an important aspect of the condoliase treatment to consider, however, for our study this was not data collected from the registries and thus was not analyzed at this point. We have included this aspect as a limitation in the discussion:

Line reads: “Also, side effects and treatment complications were not analyzed in this study and remain a critical aspect to review in future work.”

6. Longer follow up data if possible.

We concur with the reviewer that longer follow-up data would give a better impression of the long-term benefits and effects of the condoliase treatment. It is certainly our ambition to continue our work and examine its effect in longer follow-ups in a future study.

7. Plz check whether condoliase inj is more effective in Pfirrman Gr II & III degenerative stages as reported in some articles

The mentioned comparison has been made; we would refer specifically to figure 2C-D, showing the differences in VAS pain reduction for the different Pfirrmann classifications. Interestingly, however, our group A and group-N comparison actually suggest that grade IV might provide a positive prognostic indicator for treatment success. This is also discussed in lines 343-348. However, to emphasize other reports have highlighted Pfirrmann grade II and III were associated with enhanced improvement, we have added the following line in the discussion:

“However, others have reported higher rates of improvement for lower Pfirrmann classifications.[11]”

8. If possible any comparative studies of surgery vs condoliase injection to be mentioned

Condoliase injection remains a relatively new treatment and to best of our knowledge, no study has been published that compares the effectiveness of condoliase to standard microdiscectomy surgery or otherwise. However, the reviewer raises an interesting point that we believe would be good to underline. As such, we have added the following line as the closing statement of the discussion:

“Finally, future trials should aim to compare the safety and cost-efficacy profiles of condoliase intervention to standard microdiscectomy surgery to discern condoliase`s full potential in LDH treatment.”

Reviewer 2 Report

Thank you very much for the opportunity to review this interesting paper. Congratulations on your research and good luck in developing it.

Author Response

Thank you very much for the opportunity to review this interesting paper. Congratulations on your research and good luck in developing it.

We would like to sincerely thank the reviewer for using their time and expertise to review our work and are very humbled by their supportive and appreciative comments. Thank you
